# TERT Expression Induces Resistance to BRAF and MEK Inhibitors in BRAF-Mutated Melanoma In Vitro

**DOI:** 10.3390/cancers15112888

**Published:** 2023-05-24

**Authors:** Julie Delyon, Anaïs Vallet, Mélanie Bernard-Cacciarella, Isabelle Kuzniak, Coralie Reger de Moura, Baptiste Louveau, Fanélie Jouenne, Samia Mourah, Céleste Lebbé, Nicolas Dumaz

**Affiliations:** 1INSERM, U976, Team 1, Human Immunology Pathophysiology & Immunotherapy (HIPI), F-75010 Paris, France; 2Université Paris Cité, Institut de Recherche Saint Louis (IRSL), F-75010 Paris, France; 3Département de Dermatologie, Hôpital Saint Louis, AP-HP, F-75010 Paris, France; 4Département de Pharmacogénomique, Hôpital Saint Louis, AP-HP, F-75010 Paris, France

**Keywords:** melanoma, targeted therapy resistance, TERT, telomerase reverse transcriptase, TERT promoter mutation, BRAF

## Abstract

**Simple Summary:**

TERT promoter mutations are the most frequent mutations in melanoma, co-occur regularly with BRAF alterations and are associated with a poorer prognosis. Conflicting results have been published on the role of TERT promoter mutations in resistance to targeted therapy in melanoma. Our data suggest that the TERT mRNA level is associated with resistance to BRAF and MEK inhibitors and could therefore be a more reliable marker for prognosis than the promoter mutations. We showed that overexpression of TERT in a V600E-BRAF melanoma cell line drove resistance to BRAF and MEK inhibitors by a mechanism involving the reactivation of the MAPK pathway independently of telomere maintenance. Finally, we established that TERT inhibition is a therapeutic option in V600E-BRAF-mutated melanoma with acquired resistance to BRAF inhibition. Our results demonstrated the diversity of TERT biological activities in melanoma and highlight the therapeutic potential of targeting TERT in these tumors.

**Abstract:**

Because BRAF-mutated melanomas are addicted to the Mitogen Activated Protein Kinase (MAPK) pathway they show a high response rate to BRAF and MEK inhibitors. However, the clinical responses to these inhibitors are often short-lived with the rapid onset of resistance to treatment. Deciphering the molecular mechanisms driving resistance has been the subject of intense research. Recent in vitro and clinical data have suggested a link between expression of telomerase and resistance to targeted therapy in melanoma. TERT promoter mutations are the main mechanism for the continuous upregulation of telomerase in melanoma and co-occur frequently with BRAF alterations. To understand how TERT promoter mutations could be associated with resistance to targeted therapy in melanoma, we conducted translational and in vitro studies. In a cohort of V600E-BRAF-mutated melanoma patients, we showed that the TERT promoter mutation status and TERT expression tended to be associated with response to BRAF and MEK inhibitors. We demonstrated that TERT overexpression in BRAF-mutated melanoma cells reduced sensitivity to BRAF and MEK independently of TERT’s telomer maintenance activity. Interestingly, inhibition of TERT reduced growth of BRAF-mutated melanoma including resistant cells. TERT expression in melanoma can therefore be a new biomarker for resistance to MAPK inhibitors as well as a novel therapeutic target.

## 1. Introduction

Melanoma is the most aggressive skin cancer whose incidence rises every year. It can be classified on the basis of its sun exposure, anatomic site and mutational signature. Using the histopathologic degree of cumulative solar damage (CSD) of the surrounding skin, melanoma can be divided into a low-CSD group which includes superficial spreading melanoma and a high-CSD group encompassing lentigo maligna and desmoplastic melanoma. The “non-CSD” category includes spitzoid melanoma, acral melanoma and mucosal melanoma. Melanoma associated with a low level of UV radiation exposure frequently carry a *BRAF* mutation, which is present in approximately 50% of cutaneous melanomas. Meanwhile, those associated with a high level of UV radiation exposure are more likely to have a *NRAS* mutation, present in about 15% of cutaneous melanomas. The non-sun-related melanomas carry a low frequency of *KIT* mutations [1]. Because these mutations lead to activation of the MAPK pathway, several BRAF (BRAFi) and MEK (MEKi) inhibitors have been developed to treat these tumors and in particular BRAF-mutated melanoma [2]. Clinical trials have shown a high response rate to these inhibitors, but these responses are transient due to the rapid appearance of resistance to treatment. Several mechanisms of resistance to BRAF and MEK inhibitors have been described to date but the main one is MAPK reactivation followed by activation of the PI3K/AKT pathway due to *NRAS* or *MEK* mutations, BRAF splice variants or upregulation of tyrosine kinase receptors [3,4,5,6]. Recent data showed that *TERT* promoter mutations, in BRAF-mutant melanoma cell lines, were associated with sensitivity to BRAF and MEK inhibitors highlighting a link between the expression of telomerase and sensitivity to targeted therapy [7].

Telomerase is a ribonucleoprotein complex composed of a catalytic subunit, the reverse transcriptase (TERT) and an RNA template. The main function of telomerase is to maintain telomere repetitions at the ends of eukaryotic chromosomes and, hence, to preserve their integrity by preventing fusion of the chromosomal ends [8]. Telomerase is expressed during development, but its expression is repressed in most human somatic cells. However, 90% of human cancers express telomerase to maintain telomeres preventing cellular senescence and inducing immortalization of neoplastic cells [9]. The mechanisms by which the telomerase is de-repressed in tumor cells include amplification of the *TERT* gene, activation of oncogenes, such as MYC and specific mutations in the promoter of the *TERT* gene which promote the transcription of TERT. The latter is the mechanism of choice for the expression of TERT in melanoma which present a high frequency of *TERT* promoter mutations not only in sporadic melanoma (70%) but also in familial melanoma [10,11]. A total of seven *TERT* promoter mutations were identified in melanoma, the most frequent being C228T and C250T (corresponding, respectively, to −124C > T and −146C > T from the translation initiation site), creating a consensus recognition site (CCGGAA) for the transcription factors of the ETS (E26 transformation specific) family [11]. These mutations co-occur frequently with *BRAF* alterations [12] and are correlated with increased aggressiveness and poorer prognosis [13] suggesting a functional link between BRAF signaling and telomerase reactivation in melanoma.

Multiple molecular mechanisms can explain the connection between the MAPK pathway and *TERT* promoter mutation, as recently reviewed in [14]. In brief, constitutive RAS-ERK signaling can allow TERT reactivation either directly by phosphorylation of ETS1 on Thr38 by extracellular signal-regulated kinase (ERK) or indirectly by increasing GA binding protein transcription factor subunit beta (GABPB) expression and inducing a permissive promoter chromatin conformation [15]. Then, both ETS1 and GABP can bind to the ETS-binding motifs created by mutations in the *TERT* promoter and drive TERT expression. The close connection between MAPK kinase activation and TERT expression led to investigating the correlation between TERT promoter mutations and resistance to targeted therapies in melanoma. However, two studies evaluating this correlation gave conflicting results with one showing an association between *TERT* promoter mutations with longer progression-free and overall survival in patients with BRAF-mutant melanoma receiving BRAF and MEK inhibitor therapy [16]; meanwhile, the other one demonstrated that the *TERT* promoter mutation was an independent prognostic marker for the poor prognosis MAPK inhibitors-treated melanoma [13].

Hence, it is not clear whether *TERT* promoter mutations are associated with resistance to targeted therapy in melanoma. In our cohort of BRAF-mutated melanoma patients, we showed that *TERT* promoter mutation status and TERT expression tended to be associated with the response to BRAF and MEK inhibitors. Therefore, to better understand the connection between TERT expression and resistance to BRAF and MEK inhibitors, we conducted studies in BRAF-mutated melanoma cells. We showed that TERT overexpression reduced sensitivity to MAPK inhibitors independently of its telomerase activity. Finally, we demonstrated that the inhibition of TERT reduced the growth of BRAF-mutated melanoma including resistant cells.

## 2. Materials and Methods

### 2.1. Translational Studies

A total of 48 advanced or metastatic melanoma patients followed as part of routine care at Saint Louis hospital and identified in the French melanoma cohort, MelBase, were included in this retrospective study from November 2012 to November 2019. MelBase (NCT02828202) is a clinical database approved by the French Ethics Committee (CPP Ile-de-France XI, n°12027, 2012) and dedicated to the follow up of advanced and metastatic melanoma patients. Written informed consent was obtained from all patients. For each patient, baseline tumor tissues were collected, stored as formalin-fixed, paraffin-embedded (FFPE), or frozen samples and DNA was extracted as previously described [17]. mRNA was available for only 19 patients and extracted as previously described [17]. The mutational status of the TERT core promoter region (from position −27 to −286 from ATG start site) was determined by PCR and Sanger sequencing as previously described [18]. For patients with BRAF-mutated melanoma treated with BRAF and MEK inhibitors as first or following lines, progression-free survival (PFS) was computed as the time between treatment initiation and progression or death under therapy.

### 2.2. Reagents and Plasmids

Vemurafenib (V600E-BRAF inhibitor), Cobimetinib (MEK inhibitor) and 6-thio-dG (telomer capping inhibitor) were obtained from Seleckhem (Houston, TX, USA). pCDNA-3xHA-hTERT was a gift from Steven Artandi (Addgene plasmid # 51637; [19]). For stable TERT expression, melanoma cells were transfected with JetPEI (Polyplus-transfection, Illkirch, France) according to the manufacturer’s instructions and selected with G418 (10 μg/mL; Gibco, Carlsbad, CA, USA) for further analysis.

### 2.3. Cell Culture and Proliferation Assays

A375 and SkMel28 melanoma cell lines were cultured in RPMI 1640 or DMEM (Invitrogen, Cergy Pontoise, France) containing 10% (*v*/*v*) fetal calf serum (FCS; Perbio, Bredières, France), L-glutamin (2 mM; Invitrogen, Cergy Pontoise, France) and antibiotics (100 U/mL of penicillin and 1000 μg/mL of streptomycin; Invitrogen, Cergy Pontoise, France). The identity of the cell lines used in this study was confirmed by NGS. Both cell lines carry the V600E-BRAF mutation and the following mutations in the TERT promoter: −146C > T (A375) and −57C > A (SkMel28). A375^RES^ et SkMel28^RES^ were derived from A375 and SkMel28 by culture with increasing doses of vemurafenib up to 10 μM and 8 μM, respectively, as previously described [20]. The clonogenic assays and spheroid culture were previously described [20]. For proliferation assays, cells were seeded at 5 × 10^3^ cells/well into a 96-well plate in triplicate and treated with inhibitors or DMSO used as control. For EC50 measurement, cells were treated with 8 concentrations of inhibitor (from 10 μM to 0.1 nM). Proliferation was monitored using an IncuCyte^®^ Live-Cell Analysis System with repeated scanning every 3 h for 72 h (Sartorius, Göttingen, Germany). Growth inhibition and EC50 values for individual compounds were calculated using the IncuCyte^®^ software v2019B. 

### 2.4. Reverse Transcription and Real-Time PCR

Total RNA was extracted from cells cultured using Nucleospin RNA kit (MACHEREY-NAGEL GmbH & Co., KG, Duren, Germany). Reverse transcription was performed with the Go Script Reverse Transcription system (Promega, Charbonnières, France) using 1 μg of total RNA. Transcript levels were measured by qRT–PCR using SYBR green master (Applied Biosystems, Foster City, CA, USA) on a Lightcycler 480 (Roche, Indianapolis, IN, USA)). Transcript levels were normalized with the transcripts from ACTIN. The relative quantification of TERT mRNA expression was performed, comparing the Ct value of TERT to the Ct value of ACTIN using the formula 2^ΔΔCt^. The following primers were used: TERT-F: CATTTTTCCTGCGCGTCAT, TERT-R: GCGACATCCCTGCGTTCT, ACTIN-F: TGCCGACAGGATGCAGAAG and ACTIN-R: CTCAGGAGGAGCAATGATCTTGA.

### 2.5. Western Blotting

Melanoma cells were lysed using RIPA buffer supplemented with a proteinase inhibitor cocktail. Whole-cell lysates were resolved by SDS—PAGE and transferred onto nitrocellulose membranes. Membranes were probed with the following primary antibodies diluted 1 in 1000: p-ERK, ERK, (Cell Signaling Technology, Danvers, MA, USA) and TERT (ab32020 abcam). Proteins were revealed with a SuperSignal^®^ West Pico Chemiluminescent Substrate (Thermo Scientific, Rockford, IL, USA) on an ImageQuant imaging system and quantified using Image J software v1.52k (National Institutes of Health, Bethesda, MD, USA). The quantification of Western blots is presented in the Appendix A.

### 2.6. Statistical Analysis 

The mRNA expression of TERT and the PFS was considered as a continuous variable and summarized as median and IQR. Categorical variables such as the TERT promoter mutation were reported as counts and percentages. The association of variables with the TERT promoter mutation was assessed using the Mann–Whitney test. All tests were two-sided, and a *p*-value < 0.05 was considered statistically significant. GraphPad Prism v8 was used for statistical analyses in the translational study.

## 3. Results

### 3.1. TERT Promoter Mutation Status and TERT Expression Are Associated with Response to BRAF and MEK Inhibitors

Tumor samples from 48 patients with advanced BRAF-mutated melanoma were collected before treatment and analyzed for *TERT* promoter mutation status (Appendix A). The *TERT* promoter mutation was present in 43 patients (90%); 5 patients were wild type (10%). The most frequent *TERT* promoter mutation was −124C > T (n = 25, 52%), followed by −146C > T (n = 15, 31%) and −138/−139CC > TT (n = 3, 6%).

The mRNA expression of TERT was studied in the same tumor samples (n = 19 samples available; Appendix A). There was a trend towards a higher TERT expression in tumors with the mutated *TERT* promoter than in samples with the wild-type *TERT* promoter (Figure 1A, ns). 

In this cohort, 45 patients received a combination of BRAF and MEK inhibitors for metastatic BRAF-mutated melanoma. Among them, 5 (11%) patients had a wild-type *TERT* promoter status, while 24 (53%), 13 (29%) and 3 (7%) tumors had the −124C > T, the −146C > T, and the −138/−139CC > TT mutation, respectively. The median progression free survival (PFS) after starting BRAF and MEK inhibitors was 5.6 months. In patients with the wild-type TERT promoter, the median PFS was 13 months while it was 4.9, 5.3 and 3.9 months in patients with −124C > T, −146C > T and −138/−139CC > TT-mutated *TERT* promoter, respectively (*p* = 0.06) (Figure 1B).

Although the association between TERT expression and the PFS could not be directly assessed in this small cohort, these results suggest that the high expression level of TERT might be involved in resistance to BRAF and MEK inhibitors in patients with BRAF-mutated melanomas. To test this hypothesis, we established an experimental in vitro model of human BRAF-mutated melanoma cell lines with acquired resistance to BRAF and MEK inhibitors. From the melanoma cell lines A375 and SkMel28, both mutated on BRAF, resistant cell lines called A375^RES^ and SkMel28^RES^ were derived by selective growth in a medium containing increasing concentrations of the BRAF inhibitor, vemurafenib. As expected, BRAFi-resistant melanoma cell lines A375^RES^ and SkMel28^RES^ showed a high proliferation rate despite treatment with vemurafenib at 2 μM or the MEK inhibitor cobimetinib at 0.4 μM while the proliferation of parental cell lines was strongly inhibited (Figure 2A). The EC50 of vemurafenib and cobimetinib for the proliferation of parental and resistant cell lines was determined. A375^RES^ and SkMel28^RES^ showed, respectively, a 20- to 32-fold increased resistance to vemurafenib and a 45- to 55-fold increased resistance to cobimetinib in comparison to parental cell lines (Figure 2B). To evaluate whether resistance was associated with an increased TERT expression, the TERT mRNA expression was measured by RT-qPCR in parental and resistant cells. The mRNA of expression of TERT was increased significantly in A375^RES^ and SkMel28^RES^ as compared to parental cells (Figure 2C).

### 3.2. TERT Expression Is Associated with Decreased Sensitivity to BRAF and MEK Inhibition in BRAF-Mutated Melanoma Cell Lines

To study the role of TERT in resistance to BRAFi and MEKi in BRAF-mutated melanoma, we overexpressed TERT in the BRAF-mutated melanoma cell line SkMel28, which expressed a low level of TERT due to a mutation at −57 bp in the *TERT* promoter (Figure 2C). Two pools of cells overexpressing TERT were established: SkMel28-TERT1 and SkMel28-TERT2. The overexpression of TERT at the protein level in SkMel28-TERT1 and SkMel28-TERT2 compared to the parental SkMel28 was confirmed by Western blot (Figure 3A). We tested the effect of TERT overexpression on clone formation in the presence of vemurafenib at 3 μM or 10 μM. The colony formation of SkMel28 was reduced by 92% with 3 μM vemurafenib while SkMel28 overexpressing TERT had reduced sensitivity to BRAF inhibition at 3 μM, with only a 10% decrease in colony formation (Figure 3B). The 10 μM vemurafenib inhibited completely the clone formation in parental cells whereas some clones could still be detected in SkMel28 overexpressing TERT (Figure 3B). We further studied the effect of BRAFi and MEKi on the proliferation of SkMel28 and SkMel28 overexpressing TERT. We found that SkMel28-TERT1 and SkMel28-TERT2 had reduced sensitivity to vemurafenib and cobimetinib as compared to SkMel28 (Figure 3C). This result prompted us to measure the EC50 of vemurafenib and cobimetinib for the proliferation of parental and TERT overexpressing cells. In comparison with the SkMel28 parental cell line, we showed a 31- to 80-fold increased resistance to vemurafenib for SkMel28-TERT1 and SkMel28-TERT2, respectively, and a 20- to 55-fold increased resistance to cobimetinib for SkMel28-TERT1 and SkMel28-TERT2, respectively (Figure 3D), suggesting that TERT could be involved in resistance to both BRAF and MEK inhibition. To understand the mechanism of resistance to MAPK inhibition, SkMel28, SkMel28^RES^, SkMel28-TERT1 and SkMel28-TERT2 were treated with 1 μM Vemurafenib and the effect on ERK phosphorylation was evaluated by Western blot. Western blot analyses revealed that ERK phosphorylation was strongly reduced in parental SkMel28 treated with vemurafenib 1 μM. However, in SkMel28 overexpressing TERT, ERK phosphorylation was maintained despite treatment with vemurafenib 1 μΜ similar to what we observed in the resistant SkMel28^RES^ cells (Figure 3E). Similar results were obtained when cells were treated with cobimetinib (Appendix A). 

Altogether, these results suggest that the overexpression of TERT was associated with increased resistance to BRAF and MEK inhibition in BRAF-mutated melanoma cell lines by a mechanism involving the reactivation of the MAPK pathway. To evaluate whether the TERT telomerase activity was necessary for MAPK pathway reactivation, the cells overexpressing TERT were treated with vemurafenib 1 μM or cobimetinib 0.1 μM in the presence of 6-thio-dG-2′-deoxyguanosie (6-thio-dG). 6-thio-dG is a telomerase substrate precursor that is rapidly incorporated into the telomeres of cells expressing telomerase, acting as an uncapping agent and leading to a rapid induction of telomere dysfunction-induced foci [21]. The treatment with 6-thio-dG did not reduce ERK activation in cells treated with BRAFi or MEKi (Appendix A) suggesting that the effect of TERT overexpression on MAPK pathway reactivation is independent of its telomerase activity.

### 3.3. TERT Inhibition Is a Therapeutic Option in BRAF-Mutated Melanoma with Acquired Resistance to BRAF Inhibition In Vitro

To study the effect of TERT inhibition on BRAF-mutated melanoma cells in culture, we treated sensitive and resistant cells with 6-thio-dG alone or in combination with vemurafenib and analyzed the effect on cellular proliferation. As expected, the proliferation of A375 and SkMel28 cell lines was strongly reduced with vemurafenib but also with 6-thio-dG with a dose-ranging effect reaching 79% and 76% inhibition at 10 μM 6-thio-dG for A375 and SkMel28 respectively (Figure 4A). In A375^RES^ SkMel28^RES^ resistant cell lines, the proliferation was decreased by TERT inhibition with 6-thio-dG reaching 63% inhibition at 10 μM 6-thio-dG for both cell lines (Figure 4A).

As we had demonstrated that TERT was involved in the reactivation of the MAPK pathway in BRAF-mutated melanoma cell line, we hypothesized that the combination of 6-thio-dG with BRAF inhibitor could have some additional inhibitory effects. The proliferation rate of A375^RES^ and SkMel28^RES^ was decreased by around 30% with vemurafenib at 1 μM, 60% with 6-thio-dG at 10 μM and almost 90% with the combination of vemurafenib at 1 μM with 6-thio-dG at 10 μM, suggesting an additional or synergistic effect of these inhibitors (Figure 4A).

We used spheroids as a 3D model to uphold the effects of TERT inhibition in a model more representative of the growth of tumors in vivo than cells grown as monolayers. Parental and resistant SkMel28 grown as melanospheres were treated with 6-thio-dG alone or in combination with vemurafenib. We showed that the sphere formation of parental as well as resistant SkMel28 cells was significantly reduced by 6-thio-dG monotherapy as well as in combination with vemurafenib (Figure 4B). Finally, we evaluated whether SkMel28 overexpressing TERT were sensitive to the telomerase inhibitor 6-thio-dG and showed that 6-thio-dG significantly reduced the proliferation of both cell lines (Figure 4C).

## 4. Discussion

Resistance to therapies is a major challenge in the management of BRAF-mutated metastatic melanoma treated with BRAFi and MEKi. The lack of response and poor outcome are often associated with genetic alterations present at baseline triggering MAPK pathway reactivation. Amongst these alterations, the role of the frequent mutations in the promoter of *TERT* has been the subject of recent investigations. Mutations in the promoter of *TERT* are probably the most frequent mutation in melanoma and have been collectively associated with more aggressive melanomas and poorer outcomes, suggesting that these alterations were a poor prognostic factor [13]. In a recent metanalysis, including 19 studies, the mutated *TERT* promoter was associated with a significantly worse overall survival (hazard ratio 1.43, 95% CI 1.05–1.95), suggesting a major role for telomerase in malignant tumors [22]. *TERT* promoter mutations are also associated with *BRAF* mutations and the co-occurrence of both genetic alterations are associated with a poorer prognosis of disease. In accordance, we found in our cohort of BRAF-mutated patients that most melanomas (85%) presented a mutation in the TERT promoter, the most frequent being −124C > T (50%) followed by −146C > T (28%) and by −138/−139CC > TT (7%). The frequent co-occurrence of *TERT* and *BRAF* alterations can be explained because *TERT* promoter mutations create binding sites for GABP and ETS1 transcription factors, which are both targets of the MAPK pathway. On one hand, ERK phosphorylates ETS1 on T38 which is required for its transcriptional activity, and on the other hand it phosphorylates and activates FOS, which activates the *GABPB* promoter, increasing the expression of GABPβ and driving the formation of the GABPα-GABPβ complex [15]. Therefore, TERT mRNA expression should be higher in tumor tissues when the *TERT* promoter and *BRAF* mutations coexist [23]. In agreement, we showed that TERT expression tended to be higher in patients with mutated TERT promoter than in patients with wild-type TERT promoter (Figure 1A), although this result was not significant in our cohort probably in part because of the sample size and the large proportion of mutated promoter.

While the link between mutations of *TERT*, *BRAF* and a poor prognosis is quite clear, the role of *TERT* alterations in resistance to BRAFi is still controversial as opposing results have been published. In our cohort of patients treated with a combination of BRAF and MEK inhibitors, we showed that the median PFS under BRAF and MEK inhibitors treatment was higher in patients with the wild-type TERT promoter than in patients with the mutated TERT promoter (Figure 1B), suggesting that the high expression level of TERT induced by mutations in the promoter might be involved in resistance to BRAF and MEK inhibitors in patients with BRAF-mutated melanomas.

These results are in agreement with the data published by Blateau et al. who demonstrated that *TERT* Promoter Mutation was an independent prognostic marker for a poor prognosis in MAPK inhibitors-treated melanoma [13]. However, they differ from the results published by Thiemann et al. who showed that *TERT* promoter mutations were associated with a longer progression-free and overall survival in patients with BRAF-mutant melanoma receiving BRAF and MEK inhibitor therapy [16]. The specific effects associated with the different *TERT* promoter mutations could explain these discrepancies. Although this is still debated, Del Bianco et al. recently showed that in a cohort of BRAF-mutated melanoma patients who received MAPK inhibitors, those with the −146C > T mutation showed a significantly worse PFS compared to those carrying the −124C > T mutation and a two-fold increased risk of progression [24]. Moreover, the effect of the *TERT* mutations on survival may be modulated by the presence of the single nucleotide polymorphism (SNP) rs2853669 which can disrupt an ETS binding site at −245 bp in the *TERT* promoter region and, hence, neutralize the effects of the *TERT* promoter mutations [25]. The authors showed that the negative prognostic effect of the *TERT* promoter mutations in melanoma patients was only visible in patients who did not carry the rs2853669 SNP [25]. Due to the small group sizes, we could not test the effects of the different *TERT* mutations or the presence of SNP or PFS in our cohort. However, we hypothesize that the TERT mRNA level may be a more reliable marker for prognosis than the promoter mutations due to the specific effects of the different mutations and their modulation by the SNP.

To further understand the effect of TERT overexpression on resistance, we overexpressed TERT in a melanoma cell line expressing a low level of TERT and showed that it was associated with increased resistance to BRAF and MEK inhibition. We further showed that TERT overexpression prevented ERK inhibition by BRAFi and MEKi and that this effect was not inhibited by the uncapping agent 6-thio-dG, demonstrating that the mechanism involving the reactivation of the MAPK pathway was independent of the telomere lengthening function of TERT. TERT has been shown to exhibit multiple biological activities, independently of its role in telomere maintenance, acting as a transcriptional regulator modulating the expression of genes in several signal pathways implicated in the hallmarks of cancer [8,26]. TERT functions as a modulator of transcription downstream of the Wnt/b-catenin pathway by forming a complex with the transcription factor BRG1 to amplify its transcriptional regulation of downstream genes [27]. TERT can also interact with NF-κB p65 to activate NF-κB target genes [28]. Independently of its transcriptional regulation, TERT contains a mitochondrial localization signal and can thus modulate mitochondrial function such as programmed cell death by acting on the pro-apoptotic factor BAX or buffering Reactive Oxygen Species (ROS) [9]. It is therefore clear that TERT expression can contribute to resistance to targeted therapies independently of its telomere maintenance activity; however, further studies are necessary to determine which pathways are modulated by TERT in human melanoma. 

Strikingly, Tan et al. showed that BRAF-mutated melanoma cell lines were more susceptible to the apoptotic effects of BRAF and MEK inhibitors when they carried a TERT promoter mutation compared to cell lines with wild-type *TERT* [7]. Unfortunately, the expression level of TERT was not evaluated in the cell lines. Although these results are in apparent contradiction to the bad prognosis associated with TERT mutations, this discrepancy could be explained by the considerations developed above on the different *TERT* mutations and the SNP, and confirms the importance of evaluating the TERT level instead of relying solely on the presence of the promoter mutation.

Finally, we demonstrated that inhibition of the telomere lengthening function of TERT reduced the proliferation of parental and resistant melanoma cells when cells were grown in 2D or as spheres known to be enriched in cells with characteristics of tumor-initiated cells. TERT overexpressing melanoma cells were also inhibited by 6-thio-dG confirming that TERT contributes to melanoma resistance through its telomerase-dependent as well as independent functions. These results demonstrated that a treatment targeting TERT is a therapeutic option in BRAF-mutated melanomas, including those resistant to BRAF and MEK inhibitors.

## 5. Conclusions

The future treatment of melanoma will use a combination of inhibitors targeting the MAPK pathway and other signaling pathways that are important for the development of melanoma. We identified TERT expression in melanoma as a new biomarker as well as a novel therapeutic target, which not only could cooperate with inhibitors of the MAPK pathway but also treat melanoma resistant to inhibitors of MAPK pathway.

## Figures and Tables

**Figure 1 cancers-15-02888-f001:**
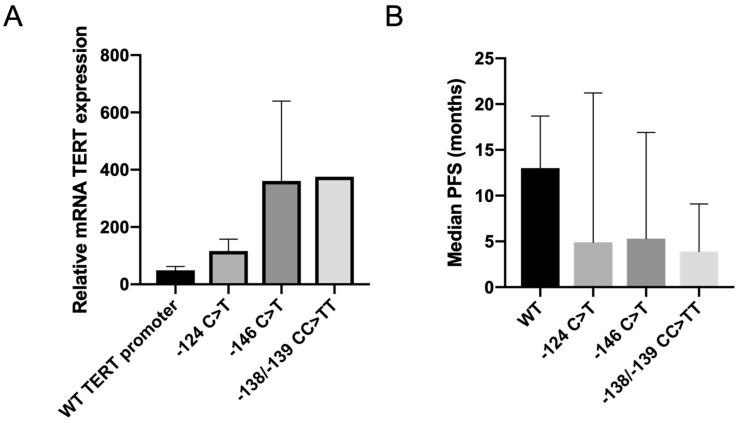
*TERT* promoter mutation status and TERT expression are associated with response to BRAF and MEK inhibitors. (**A**) Relative mRNA TERT expression (TERT/ACTIN) according to the *TERT* promoter mutation status in tumor samples. Mean and SEM (ns). (**B**) Progression-free survival (months) in advanced BRAF-mutated melanoma patients treated with BRAF and MEK inhibitors, according to *TERT* promoter mutation status. Median ± IQR (ns).

**Figure 2 cancers-15-02888-f002:**
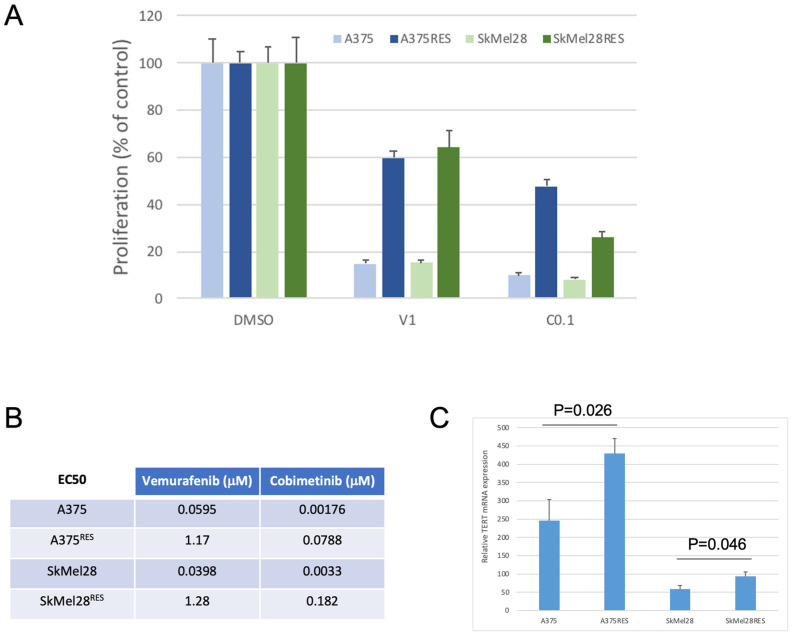
TERT expression in melanoma cell lines resistant to BRAF and MEK inhibitors. (**A**) Parental (A375 and SkMel28) and resistant (A375RES and SkMel28RES) cells were treated with DMSO, 1 μM of vemurafenib (V1) or 0.10 μM of cobimetinib (C0.1) and the proliferation was analyzed after 3 days (data are represented as mean ± s.d.). Both inhibitors induced a significant inhibition of proliferation in the four cell lines (*p* < 0.001; unpaired *t*-test). (**B**) The half maximal effective concentration (EC50) for vemurafenib and cobimetinib was measured for the cell line indicated. (**C**) Relative mRNA expression of TERT was assessed by qPCR and normalized to ACTIN in the cell line indicated (data are represented as mean ± s.d.; unpaired *t*-test).

**Figure 3 cancers-15-02888-f003:**
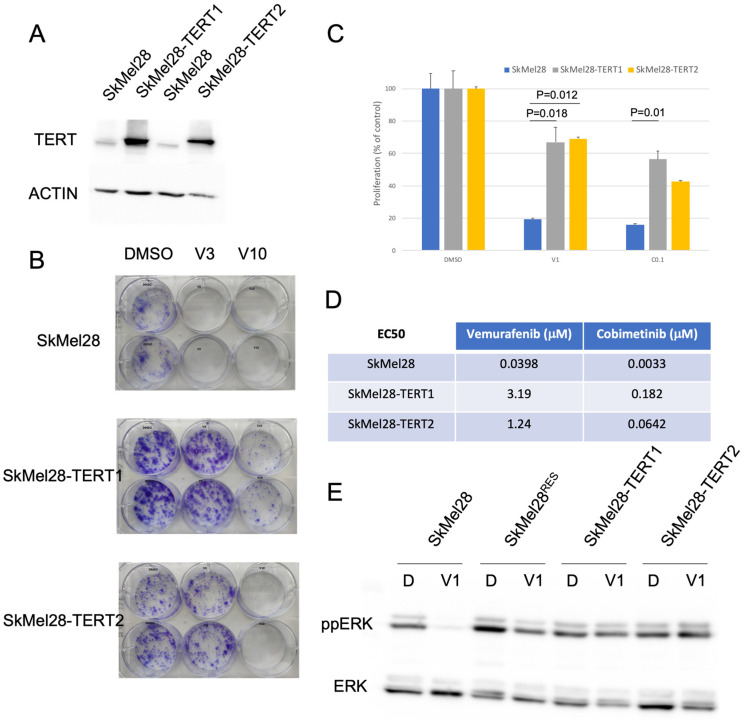
TERT over-expression is associated with decreased sensitivity to BRAF and MEK inhibition. (**A**) SkMel28 cells were transfected with a vector expressing human TERT. The expression of TERT and ACTIN was assessed by Western blotting. (**B**) SkMel28 parental and overexpressing TERT were seeded at low density and treated three times a week with DMSO, Vemurafenib 3 μM (V3) or 10 μM (V10) and fixed and stained after 2 weeks. (**C**) The same cell lines were treated with DMSO, 1 μM of vemurafenib (V1) or 0.10 μM of cobimetinib (C0.1) and the proliferation was analyzed after 3 days (data are represented as mean ± s.d.; unpaired *t*-test). (**D**) The EC50 for vemurafenib and cobimetinib was measured for the cell line indicated. (**E**) The indicated cells were treated for 24 h with DMSO or vemurafenib 1 μM (V1) and the expressions of phosphorylated ERK (ppERK) and total ERK (ERK) were assessed by Western blotting. Original blots see Appendix A.

**Figure 4 cancers-15-02888-f004:**
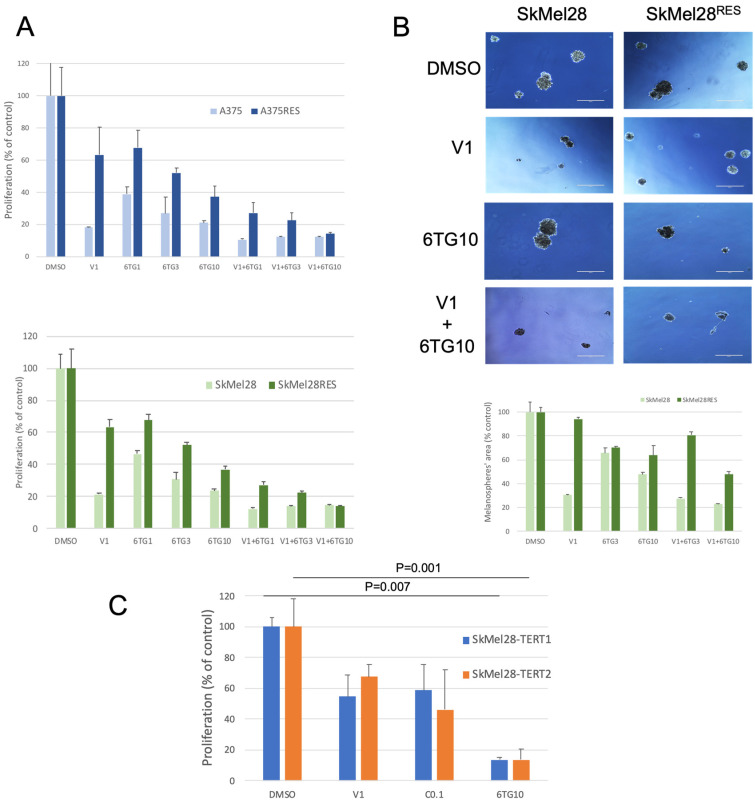
TERT inhibition reduces proliferation of BRAF-mutated melanoma with acquired resistance to BRAF inhibition. (**A**) Parental and resistant A375 and SkMel28 were treated with DMSO, 1 μM of vemurafenib (V1), 1 μM (6TG1), 3 μM (6TG3) and 10 μM (6TG10) of 6-thio-dG or a combination of vemurafenib and 6-thio-dG and the proliferation was analyzed after 3 days (data are represented as mean ± s.d.). All inhibitors induced a significant inhibition of proliferation in the four cell lines (*p* < 0.001; unpaired *t*-test). (**B**) Parental and resistant SkMel28 were grown as spheroids and treated with DMSO, 1 μM of vemurafenib (V1), 10 μM of 6-thio-dG (6TG10) or a combination of both for 10 days. Pictures were taken after 10 days of treatment; scale bar represents 1000 μm. Graphs represent the mean melanospheres’ area ± s.d. All inhibitors induced a significant inhibition of proliferation in both cell lines (*p* < 0.0001; unpaired *t*-test) except V1 in SkMel28^RES^. (**C**) SkMel28 overexpressing TERT were treated with DMSO, 1 μM of vemurafenib (V1), 0.1 μM of cobimetinib (C0.1) or 10 μM of 6-thio-dG (6TG10) and the proliferation was analyzed after 3 days (data are represented as mean ± s.d.; unpaired *t*-test).

## Data Availability

The data presented in this study are available in this article (and Appendix A).

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
