# Peer review of "TERT Expression Induces Resistance to BRAF and MEK Inhibitors in BRAF-Mutated Melanoma In Vitro"

_cancers, 2023, doi:10.3390/cancers15112888_

Round 1
Reviewer 1 Report
I thank the academic editor for giving me the opportunity to review this manuscript in which the authors conduct a translational study with melanoma cell lines to investigate the impact of TERT promoter mutations in malignant melanoma, based on a strong interest in this mechanism molecule present in the literature. I believe that the materials and methods, results and discussion address this topic well and I can suggest the authors to add a paragraph related to the histopathological diagnosis of melanoma, so that we can implement and improve the manuscript.Moderate change
Author Response
I thank the academic editor for giving me the opportunity to review this manuscript in which the authors conduct a translational study with melanoma cell lines to investigate the impact of TERT promoter mutations in malignant melanoma, based on a strong interest in this mechanism molecule present in the literature. I believe that the materials and methods, results and discussion address this topic well and I can suggest the authors to add a paragraph related to the histopathological diagnosis of melanoma, so that we can implement and improve the manuscript.
Following the reviewer's request, we added, in the introduction, the following paragraph related to the histopathological diagnosis of melanoma :
Melanoma is the most aggressive skin cancer whose incidence rises every year. it can be classified on the basis of its sun exposure, anatomic site and mutational signature. Using the histopathologic degree of cumulative solar damage (CSD) of the surrounding skin, melanoma can be divided into a low-CSD group which includes superficial spreading melanoma and a high-CSD group encompassing lentigo maligna and desmoplastic melanoma. The "non-CSD" category includes spitzoid melanoma, acral melanoma and mucosal melanoma. Melanoma associated with of low level of UV radiation exposure frequently carry a BRAF mutation, which is present in approximately 50% of cutaneous melanomas. Whereas those associated with high level of UV radiation exposure are more likely to have a NRAS mutation, present in about 15% of cutaneous melanomas. The non-sun-related melanomas carry a low frequency of KIT mutations [1]. Because these mutations lead to activation of the Mitogen Activated Protein Kinase (MAPK) pathway, several BRAF (BRAFi) and MEK (MEKi) inhibitors have been developed to treat these tumors and in particular BRAF mutated melanoma [2]. Clinical trials have shown a high response rate to these inhibitors, but these responses are transient due to the rapid appearance of resistance to treatment. Several mechanisms of resistance to BRAF and MEK inhibitors have been described to date but the main one is MAPK reactivation followed by activation of the PI3K/AKT pathway due to NRAS or MEK mutations, BRAF splice variants or upregulation of tyrosine kinase receptors [3-6]. Recent data showed that TERT promoter mutations, in BRAF-mutant melanoma cell lines, were associated with sensitivity to BRAF and MEK inhibitors highlighting a link between expression of telomerase and sensitivity to targeted therapy [7].
Reviewer 2 Report
In the manuscript entitled “TERT expression induces resistance to BRAF and MEK inhibitors in BRAF-mutated melanoma” the authors showed that overexpression of TERT in a V600E-BRAF melanoma cell line drove resistance to BRAF and MEK inhibitors by a mechanism involving the reactivation of the MAPK pathway independently of telomere maintenance. Finally, they highlighted that TERT inhibition may serve as a therapeutic option in V600E-BRAF-mutated melanoma with acquired resistance to BRAF inhibition. Their results also demonstrated the diversity of TERT biological activities in melanoma and highlighted the therapeutic potential of targeting TERT in these tumors. The manuscript is valuable in general, however some concerns have to be taken into account before publication:
Translational studies. Number of patients enrolled
How much RNA was transcribed
Ab dilution
Line 166: why 19?
Fig.1A: SEM in the last histogram CC>TT
Fig.2A: proliferation for 24h and 48h? how many replicates?
Fig.2C: arbitrary unit??
Line 214: probably due to a mutation at -57 bp??? Confirm by sequencing?
Line 221 and line 233: why 1µM?
Line 366: ref.7, also in other types of cancer (10.3390/ijms23116183)
N/A
Author Response
Translational studies. Number of patients enrolled
48 patients were enrolled in the translational study but mRNA was available for only 19 patients. This is now indicated in Materials and Methods.
How much RNA was transcribed
1μg of total RNA was used in the reverse transcription. This is now indicated in Materials and Methods.
Ab dilution
The antibody dilutions are now indicated in Materials and Methods.
Line 166: why 19?
48 patients were enrolled in the translational study but mRNA was available for only 19 patients. This is now indicated in Materials and Methods and Results.
Fig.1A: SEM in the last histogram CC>TT
Due to the rarity of this mutation, it was only found in 1 out of the 19 patients for which mRNA was available, hence the lack of SEM.
Fig.2A: proliferation for 24h and 48h? how many replicates?
Details of the proliferation assays are now indicated in Materials and Methods: For proliferation assays, cells were seeded at 5.103 cells/well into a 96-well plate in triplicate and treated with inhibitors or DMSO used as control. For EC50 measurement, cells were treated with 8 concentrations of inhibitor (10μM to 0.1nM). Proliferation was monitored using an IncuCyte® Live-Cell Analysis System with repeated scanning every 3 hours for 72 hours (Sartorius, Germany.) Growth inhibition and EC50 values for individual compounds were calculated using the IncuCyte® software. Although proliferation was monitored regularly during 72 hrs (see figure below), we chose to present the data at 72 hrs because cells were in their exponential growth phase. 
Fig.2C: arbitrary unit??
The relative quantification of TERT mRNA expression was done comparing the Ct value of TERT to the Ct value of ACTIN using the formula 2ΔΔCt.This is now indicated in Materials and Methods and the "arbitrary unit" has been removed.
Line 214: probably due to a mutation at -57 bp??? Confirm by sequencing?
We apologize for the confusing sentence. The TERT promoter of SkMel28 was sequenced and found to carry the mutation at -57bp. We meant to say that the low level of TERT expression in this cell line is probably linked to the fact that this cell line carries the rare -57 bp mutation known to be a weak activator of TERT expression . The sentence was corrected.
Line 221 and line 233: why 1µM?
We chose to treat cells with 1µM vemurafenib for 24 hrs because this dose strongly inhibits ERK phosphorylation in the parental cells without killing the cells in 24 hrs, allowing us to recover enough proteins for analysis.
Line 366: ref.7, also in other types of cancer (10.3390/ijms23116183)
This reference was added to the discussion.
Reviewer 3 Report
Reviewed article „TERT expression induces resistance to BRAF and MEK inhibitors in BRAF-mutated melanoma” is extensive, a comprehensive introduction that gives a full insight into the subject, and a well-written discussion, trying to explain the emerging discrepancies and similarities between own and other authors' results. Interesting results of extensive research, although it can be said that it is still preliminary. The authors themselves note a relatively small group of the studied cohort, or the lack of SNP results. However, the results obtained in this study provide a lot of information about the relationship between TERT and the treatment of melanoma with BRAF/MEK inhibitors.
However further studies are necessary to determine which pathways are modulated by TERT in human melanoma. In addition, due to the problematic treatment of melanoma, it is important to know the exact mechanisms of carcinogenesis and to search for new concepts of therapy, therefore, I consider the presented results to be significant.
However, I have a few comments and questions about the manuscript:
1) In „matherials and methods” in point „Cell culture” there is information about obtaining a RES line by culture with vemurafenib up to 10 μM and 8 μM. Where did these doses come from?? It is worth justifying it or providing a reference to the research results presented below.
2) Why did the "statistical analysis" show the results as "median and IQR" and not "median and SEM or SD"? It would be worth briefly justifying this.
3) In the description of the results in line 162 there is a reference to "(supplementary table 1)", but in the supplementary materials there is no mentioned table.
4) In line 161 we find out that the material was taken from 48 patients, and in line 163 that 43 had TERT promoter mutation. So why in the next line number 166 "mRNA expression of TERT" was only evaluated in 19 patients/19 samples??
5) Lines 193-194 and 227. What method was used to determine EC50 doses of vemurafenib and cobimetinib for the proliferation of cell lines?? From how many intermediate values, the ES50 dose was determined??
Author Response
1) In „matherials and methods” in point „Cell culture” there is information about obtaining a RES line by culture with vemurafenib up to 10 μM and 8 μM. Where did these doses come from?? It is worth justifying it or providing a reference to the research results presented below.
The resistant cells were previously described, the reference was added in Materials and Methods.
2) Why did the "statistical analysis" show the results as "median and IQR" and not "median and SEM or SD"? It would be worth briefly justifying this.
IQR was chosen because there was a large distribution of PFS in the cohort with extreme values (2 or 73 months for instance). The following sentence was added in the statistical section: “The mRNA expression of TERT and the PFS was considered as continuous variable and summarized as median and IQR”.
3) In the description of the results in line 162 there is a reference to "(supplementary table 1)", but in the supplementary materials there is no mentioned table.
The supplementary table 1 presents the tumor samples' characteristics. It was originally uploaded separately from the supplementary materials but is now included in the same file.
4) In line 161 we find out that the material was taken from 48 patients, and in line 163 that 43 had TERT promoter mutation. So why in the next line number 166 "mRNA expression of TERT" was only evaluated in 19 patients/19 samples??
48 patients were enrolled in the translational study but mRNA was available for only 19 patients. This is now indicated in Materials and Methods and Results.
5) Lines 193-194 and 227. What method was used to determine EC50 doses of vemurafenib and cobimetinib for the proliferation of cell lines?? From how many intermediate values, the ES50 dose was determined??
For proliferation assays, cells were seeded at 5.103 cells/well into a 96-well plate in triplicate and treated with inhibitors or DMSO used as control. For EC50 measurement, cells were treated with 8 concentrations of inhibitor (10μM to 0.1nM). Proliferation was monitored using an IncuCyte® Live-Cell Analysis System with repeated scanning every 3 hours for 72 hours (Sartorius, Germany.) Growth inhibition and EC50 values for individual compounds were calculated using the IncuCyte® software. This is now indicated in Materials and Methods.
